# The Yeast Protein Kinase Sch9 Functions as a Central Nutrient-Responsive Hub That Calibrates Metabolic and Stress-Related Responses

**DOI:** 10.3390/jof9080787

**Published:** 2023-07-26

**Authors:** Marco Caligaris, Belém Sampaio-Marques, Riko Hatakeyama, Benjamin Pillet, Paula Ludovico, Claudio De Virgilio, Joris Winderickx, Raffaele Nicastro

**Affiliations:** 1Department of Biology, University of Fribourg, 1700 Fribourg, Switzerland; marco.caligaris@unifr.ch (M.C.); benjamin.pillet@unifr.ch (B.P.); claudio.devirgilio@unifr.ch (C.D.V.); 2Life and Health Sciences Research Institute (ICVS), School of Medicine, University of Minho, 4710-057 Braga, Portugal; mbmarques@med.uminho.pt (B.S.-M.); pludovico@med.uminho.pt (P.L.); 3ICVS/3B’s-PT Government Associate Laboratory, 4806-909 Guimarães, Portugal; 4Institute of Medical Sciences, University of Aberdeen, Aberdeen AB25 2ZD, UK; riko.hatakeyama@abdn.ac.uk; 5Department of Biology, Functional Biology, KU Leuven, B-3001 Heverlee, Belgium; joris.winderickx@kuleuven.be

**Keywords:** Sch9, TORC1, SNF1, Pkh1, Pkh2, Pkh3, Pho85, lipid, stress, longevity

## Abstract

Yeast cells are equipped with different nutrient signaling pathways that enable them to sense the availability of various nutrients and adjust metabolism and growth accordingly. These pathways are part of an intricate network since most of them are cross-regulated and subject to feedback regulation at different levels. In yeast, a central role is played by Sch9, a protein kinase that functions as a proximal effector of the conserved growth-regulatory TORC1 complex to mediate information on the availability of free amino acids. However, recent studies established that Sch9 is more than a TORC1-effector as its activity is tuned by several other kinases. This allows Sch9 to function as an integrator that aligns different input signals to achieve accuracy in metabolic responses and stress-related molecular adaptations. In this review, we highlight the latest findings on the structure and regulation of Sch9, as well as its role as a nutrient-responsive hub that impacts on growth and longevity of yeast cells. Given that most key players impinging on Sch9 are well-conserved, we also discuss how studies on Sch9 can be instrumental to further elucidate mechanisms underpinning healthy aging in mammalians.

## 1. Introduction

Proper coordination between nutrient availability and cell growth is needed in all organisms to guarantee a successful adaptation to environmental changes. In the budding yeast *Saccharomyces cerevisiae*, several signal transduction pathways are involved in nutrient sensing, regulation of cell growth, and cell cycle progression. Among them, the TORC1 signaling pathway plays a key role in nitrogen sensing and signaling [1,2,3,4,5]. Responding to nutrient cues allows TORC1 to promote growth by regulating catabolic and anabolic processes [6]. The TORC1 structure is well-conserved in eukaryotes and in budding yeast. It consists of a multimeric complex that consists of the serine/threonine protein kinase catalytic subunit (Tor1 or Tor2) and three regulatory subunits (Kog1, Tco89, and Lst8) [7]. By phosphorylating its main target Sch9 [8], TORC1 regulates ribosome biogenesis, translation initiation, protein synthesis, sphingolipid metabolism, cell cycle progression, cell size, stress response, and autophagy in response to nutrient availability [9,10,11,12,13,14]. *SCH9* (for Scott Cameron *Hind*III library clone number 9; Takashi Toda; personal communication) has been initially identified as a multicopy suppressor of the growth defect caused by a temperature-sensitive *cdc25* allele [15], hinting at a connection with the Ras/PKA signaling pathway. Sch9 belongs to the family of AGC protein kinases, and it is localized both in the cytosol and on vacuolar membranes [11,16,17], from where it is dispersed into the cytosol when cells are limited for carbon sources [18]. Full activation of Sch9 requires phosphorylation of different amino acid residues by various protein kinases (Figure 1a). These include the paralogous Pkh1, Pkh2, and Pkh3 kinases, which themselves are stimulated [19,20,21] or not [22] by phytosphingosine (PHS) (see [23] for a detailed discussion) and which phosphorylate Thr^570^ in the activation loop of Sch9 [8,19,24,25]. In addition, TORC1 phosphorylates multiple residues in the C-terminal region of Sch9 [8], and the cyclin-dependent protein kinases (CDKs) Bur1 and Pho85 phosphorylate Thr^723^ and Ser^726^, both originally considered TORC1 target residues [26,27]. In contrast, Sch9 can also be inhibited through phosphorylation of Ser^288^ by SNF1 when cells are starved for carbon [28,29,30]. In line with its role as a central signaling node, Sch9 regulates a large number of downstream effectors, such as the glucose-responsive protein kinase A (PKA) [31], which is encoded by the paralogous *TPK1*, *TPK2*, and *TPK3* genes [32]. Interestingly, in this context, Sch9 exhibits a target profile that partially overlaps with the one of PKA [12,33]. In addition, Sch9 is important for stress response regulation through its effects on the activity of various protein kinases and transcription factors [34,35]. Moreover, Sch9 impacts the proteasomal and autophagic degradative systems, and loss of Sch9 causes an extension of lifespan in yeast cells [36,37,38].

In this review, we recapitulate the functions of the protein kinase Sch9, which serves as a hub that integrates inputs from different pathways, to modulate growth-related adaptations to nutrient changes, including stress response, autophagy, and longevity. We will additionally focus on the regulation of Sch9 by other signaling pathways and via lipid binding. Furthermore, a specific section will be devoted to exploring how regulatory events might affect Sch9 structure and function.

## 2. Structure of Sch9 and Confirmed Phosphosites

Although the experimental structure of Sch9 remains unresolved, the release of the Alphafold2 predicted protein structure database [39,40] provides high-accuracy models (Figure 1a) that, when combined with the known structural and functional characteristics of similar kinases, such as the yeast Ypk1 or the mammalian AKT1 and S6K1, can offer valuable insights (Figure 1b; Table 1). Based on these predicted structures and previous sequence analyses [8,42], Sch9 can be divided into four main regions: the disordered N-terminal region; the C2 domain; the kinase domain; and the C-terminal region.

The regions outlined in Table 1 and illustrated in Figure 1a–c have been defined based on the predicted structure of Sch9, Ypk1, S6K1, and AKT1. The N-terminal regions (PH + linker for AKT1) extend from the start of the protein to either the C2 domain (Sch9 and Ypk1) or the kinase domain (S6K1 and AKT1). The C2 domains were defined as starting with the first amino acid of the initial predicted β-strand (β1) and ending with the last amino acid of the final β-strand (β8), forming the β-sandwich. The kinase domains comprise the kinase core and the adjacent non-catalytic regions that are structurally aligned among the four analyzed kinases. These include the highly conserved activation loops and end with the hydrophobic motifs (HM) [43]. The C-terminal regions, consisting of all the amino acids following the HM, are predicted to have substantial structural variation among these kinases.

The N-terminus of Sch9 (amino acids: 1–183) is involved in membrane binding, and its structure is poorly predicted, indicating a predominantly disordered nature. In contrast, the N-terminus of AKT1 is well-folded and forms the PH (pleckstrin homology) domain (Figure 1b rightmost panel, purple) [44], which also engages in lipid binding [45,46,47]. Additionally, The PH domain of AKT1 fulfills an autoinhibitory role by binding to the kinase domain and shielding the activation loop [44]. Interestingly, although a part of the N-terminal region of Sch9 (spanning residues 55 to 105) exhibits a low average pLDDT (predicted Local Distance Difference Test) score, this region has a low Predicted Aligned Error (PAE) relative to the kinase domain, which indicates that it is likely positioned above the activation loop. This region of Sch9 could potentially play a role analogous to the PH domain of AKT1, keeping the activation loop buried while in its inactive form. Likewise, the N-terminal regions of Ypk1 and S6K are also predicted to be in proximity to the active site and may sterically occlude it as well (Figure 1a,c).

The first well-folded domain of Sch9 is a C2 domain of type II topology, which is characterized by the positioning of the N- and C-termini at the bottom of the β-barrel. Some C2 domains have been shown to coordinate Ca^2+^ on the loops between the β-sheets, which subsequently generates a binding surface that interacts with negatively charged lipids on the top of the C2 domain [48]. In contrast to Ypk1, which possesses α-helical structures in place of those loops, Sch9 has the potential to coordinate Ca^2+^ using the following amino acids: D201 (between β1 and β2), D353 (between β5 and β6); and D386/E387 (between β7 and β8). This mechanism could potentially stabilize the membrane tethering of Sch9 or enable Sch9 to bind to membranes with varying lipid compositions, although the membrane binding capacity of this particular C2 domain is so far questionable [16] (see also the next section). Unlike a typical C2 domain, Sch9 features an extended loop between β3 and β4. Importantly, this loop hosts Ser^288^, which is phosphorylated by SNF1 to promote the inactivation of Sch9 [28]. In addition to the functionally relevant loops, the C2 domain was hypothesized to assist the N-terminus in inhibiting the kinase domain by imposing a conformational constraint due to its close proximity [49], which would explain the constitutive and TORC2-independent activity observed in Ypk1 (D242A) and Ypk2 (D239A) mutants [50,51,52,53]. These mutations are located at the interface between the C2 domain and the kinase domain, potentially weakening the interaction between the two domains. Consequently, the C2 domain could provide an additional autoinhibitory mechanism to the steric occlusion of the active site by the N-terminus.

The spatial relationship between the C2 domain and the kinase domain could be regulated by the phosphorylation of the kinase domain at amino acid Thr^737^, which activates the Sch9 [8]. This phosphorylation has also been shown to reorder the hydrophobic motif [54]. Hypothetically, the negative charge of pThr^737^ could electrostatically interact with Arg^405^, bending the linker and connecting the C2 domain to the kinase domain. This would tilt the C2 domain away from the kinase domain, allowing for a catalytically competent conformation, as modeled in Figure 1c. The equivalent residue Arg^144^ of AKT1 was reported to drastically decrease its catalytic activity when mutated to alanine [55]. However, more recent research has challenged these findings, observing no effect resulting from the R144A mutation [44]. In the context of Sch9 and Ypk1, the linker between the two domains is shorter, suggesting that this mutation could have a more significant impact. Recently, the kinase domain was found to contain phosphorylation sites for the cyclin-dependent kinases Bur1 and Pho85 as well (Figure 1a), and especially the Pho85-mediated phosphorylation at Ser^726^ was shown to prime Sch9 for its subsequent activation by TORC1 [26,27]. Hence, it is tempting to speculate that this priming could also influence the spacing of the C2 domain and the kinase domain to relieve the occlusion by the N-terminus.

## 3. Lipid-Dependent Regulation of Sch9

Sch9 localizes throughout the cytoplasm, with a significant enrichment on the cytoplasmic surface of the vacuolar membrane [8,11]. The vacuolar localization is required for its activation through phosphorylation by TORC1, which resides on the same membrane [8,26,56]. The vacuolar targeting of Sch9 is mediated by the physical interaction between its N-terminal domain (1–183 amino acid residues; hereafter, Sch9^1–183^) with the membrane lipid phosphatidylinositol 3,5-bisphosphate (PI(3,5)P_2_) [16,17,57]. As mentioned, Sch9^1–183^ is predicted to be largely disordered, unlike many other known phosphoinositide-binding domains that have defined configurations [58,59]. It is, therefore, unclear how exactly Sch9^1–183^ interacts with PI(3,5)P_2_.

Other domains of Sch9 also influence its membrane targeting. As mentioned in the previous section, Sch9 has a C2 domain (184–402 residues), a typical lipid-binding motif [60]. Surprisingly, however, the addition of this domain partially compromises the membrane recruitment of the Sch9^1–183^ fragment, suggesting rather a negative role [16]. The underlying mechanism of this inhibitory effect is so far unknown. Another clue is provided by the comparison between the localization pattern of the full-length Sch9 and that of the Sch9^1–183^ alone. While the full-length Sch9 exclusively localizes to the cytoplasm with enrichment at the vacuolar membrane, Sch9^1–183^ alone is additionally found at signaling endosomes [16]. Since PI(3,5)P_2_ is enriched on both vacuoles and signaling endosomes, this observation suggests that the non-N-terminal region of Sch9 biases the binding preference toward the vacuolar membrane. The underlying mechanism is again unknown but may involve the interaction of the 184–824 residues with other lipids or proteins residing uniquely on the vacuolar membrane. Alternatively, different physical properties of the two organelles, for example, the membrane curvature (as the vacuoles are significantly larger than signaling endosomes), may account for the altered preference. Of note, both Sch9 and AKT1 contain a motif upstream of the activation loop that closely resembles the proposed consensus sequence for the PI(3,5)P_2_ binding [41]. However, the significance of this motif for the phosphoinositide interaction of Sch9 and AKT1 remains to be established.

Membrane targeting of Sch9 is dynamically regulated in response to environmental changes. For example, glucose starvation and oxidative stress cause Sch9 to detach from the vacuolar membrane [11,18,61]. The underlying mechanisms for the regulated (de)localization of Sch9 are not fully understood, but recent observations have shown that these also involve alterations in the amount or the subcellular distribution of PI(3,5)P_2_. Indeed, evidence exists that PI(3,5)P_2_ responds to various cellular stresses and nutrient availability. Osmotic stress, for instance, stimulates PI(3,5)P_2_ production in a manner dependent on the cyclin-dependent kinase Pho85 [62,63]. Mechanistically, Pho85 activates, in association with the Pho80 cyclin, the lipid kinase complex that converts PI3P into PI(3,5)P_2_ by direct phosphorylation of the catalytic phosphatidylinositol-3-phosphate 5-kinase subunit Fab1 and its regulatory subunit Vac7 [63]. Moreover, TORC1 phosphorylates Fab1 as well, thereby stimulating PI(3,5)P_2_ production on signaling endosomes [16]. Upon fusion of the signaling endosome to the vacuole, PI(3,5)P_2_ is delivered to the vacuolar membrane, which is required to recruit TORC1 and its substrate Sch9 [16,26,57]. Interestingly, TORC1 and Fab1 form a positive feedback loop [16]. These facts indicate that PI(3,5)P_2_ acts as an important signaling lipid that links nutrient/stress-responsive pathways, such as Pho85 and TORC1 pathways, to downstream effectors, including Sch9 [26]. Notably, the research on PI(3,5)P_2_ has been hampered by the lack of visualization tools. The use of GFP-fused Sch9^1–183^ as a PI(3,5)P_2_ biosensor may therefore be helpful in advancing the research in this area [16].

Another vacuolar membrane client that requires PI(3,5)P_2_ and that influences the activation of Sch9 is the V-ATPase, the vacuolar proton pump, whose assembly and activity are dependent on glucose availability [64,65]. Initial studies suggested that the V-ATPase plays an important role in cellular stress responses by promoting the activities of the PKA and TORC1 pathways through an interaction with the Arf1 and Gtr1 GTPases, respectively [66,67]. The connection between V-ATPase and TORC1 activation was recently proposed to be mediated by the Ccr4-Not complex, which is a known downstream effector of TORC1 for ribosomal RNA biogenesis and transcription of stress-responsive genes [68,69], but which also acts upstream of TORC1 as a regulator of V-ATPase assembly and vacuolar acidification [70]. Interestingly, Sch9 is known to influence the activity of the V-ATPase as well since it facilitates V-ATPase disassembly upon glucose starvation, thus providing a possible feedback route [18]. How Ccr4-Not and Sch9 control the V-ATPase assembly/disassembly state is not known, but one suggested possibility involves the ubiquitylation and stability of one or more V-ATPase subunits [70,71]. In addition, the V-ATPase was proposed to act as a sensor of cytosolic pH [72]. This reconciles with data showing that also proton influx at the plasma membrane, which is catalyzed by amino-acid/H^+^ symporters and driven by the H^+^-ATPase Pma1, influences the TORC1 activation [73]. Since Sch9 is required to maintain normal Pma1 activity and extracellular acidification [18], it likely influences this plasma membrane symport of protons and amino acids as well.

Besides being recruited to the vacuolar membrane via phosphoinositide PI(3,5)P_2_-binding, the Sch9 function is also fine-tuned by the long chain base (LCB) phytosphingosine (PHS), an intermediate of the sphingolipid metabolic pathway [19,74]. Here, a key role is played by protein kinases Pkh1, Pkh2, and Pkh3, the orthologues of the mammalian 3-phosphoinositide dependent kinase-1, PDK1, which are known to be involved in the maintenance of cell wall integrity and the control of eisosome dynamics [21,23,75]. These PDK1 orthologs phosphorylate the Sch9 activation loop at Thr^570^ (Figure 1a), an event that occurs independently of the C-terminal Sch9 phosphorylation by TORC1 and that is required to obtain a full Sch9 activity [8,25]. Interestingly, the Pkh-Sch9 axis appears to establish a feedback loop since, as depicted in Figure 2, sphingolipid metabolism is itself regulated by Sch9 at the level of the ceramide synthases Lac1 and Lag1, the ceramidases Ydc1 and Ypc1, as well as the inositol phosphosphingolipid phospholipase C, Isc1 [76]. The latter translocates from the endoplasmic reticulum to mitochondria during the diauxic shift and hydrolyzes the complex sphingolipids IPC, MIPC, and M(IP)_2_C back into dihydro-/phytoceramides, which contribute to the normal functioning of mitochondria [77]. This Isc1 translocation to mitochondria is dependent on Sch9, explaining at least in part the requirement of Sch9 to properly traverse the diauxic shift [76,78].

Sphingolipids are important components of membranes that, beyond their structural role, also fulfill additional specific functions in several fundamental cellular processes. For instance, the dynamic balance between the different sphingolipid metabolites, especially LCBs, their phosphorylated derivatives (LCBPs), ceramides, and complex sphingolipids, have been shown to accompany stress responses, mitochondrial functioning and oxidative phosphorylation, cell wall synthesis and repair, autophagy, endocytosis, and actin cytoskeleton dynamics, thereby affecting the growth and longevity of yeast cells [74,79,80]. In general, LCBPs have been shown to act as pro-growth signals, while ceramides mainly act as antiproliferative signals [81]. The role of complex sphingolipids is less well-understood since they appear to be dispensable for yeast cell survival. Nonetheless, IPC has been associated with the regulation of cellular Ca^2+^ homeostasis [82,83] and autophagy [80]. As compared to wild-type cells, *sch9Δ* mutant cells display enhanced levels of the long-chain bases PHS and dihydrosphingosine (DHS) and their phosphorylated derivatives, decreased levels of several (phyto)ceramide species, and altered ratios of complex sphingolipids, a profile that is believed to contribute to the increased chronological lifespan of the mutant cells [76]. However, the relationship between sphingolipid metabolism and longevity is not straightforward, and other factors are at play as well. One such factor is Sit4, the catalytic subunit of a PP2A-type protein phosphatase that is down-regulated by TORC1 but up-regulated by ceramides [84,85]. Besides Sch9, there are also other kinases targeted by the yeast PDK1 orthologues to regulate sphingolipid metabolism. Indeed, Pkh1 and Pkh2 also control the activity of the protein kinases Ypk1 and Ypk2, which upon heat shock, boost the de novo biosynthesis of sphingoid bases by phosphorylating and relieving the inhibition exerted by the two ER-localized tetraspanins Orm1 and Orm2 [86]. In addition, Ypk1 promotes the production of complex sphingolipids through activation of the Lac1 and Lag1 ceramide synthases [87]. Hence, Sch9 and Ypk1/2 share common targets to regulate sphingolipid homeostasis. Full activation of Sch9 requires TORC1 at the vacuolar membrane to signal nutrient availability. Instead, the full activation of Ypk1 and Ypk2 depends on the phosphorylation in their hydrophobic motif by the TORC2 complex, which localizes at the plasma membrane and signals membrane perturbation and stress [23,51]. Thus, Sch9, Ypk1, and Ypk2 also share a similar mode of activation. Finally, Pho85 and SNF1 were also shown to be involved in the regulation of sphingolipid metabolism. Pho85, together with one of the redundant cyclins Pcl1 and Pcl2, phosphorylates the long-chain base kinase Lcb4 thereby marking this kinase for degradation [88]. Consistently, *pho85Δ* cells are characterized by reduced LCB levels and markedly increased LCBP levels [89]. An exact target for SNF1 has not been determined, but a strain lacking the catalytic subunit Snf1 was shown to display significantly increased IPC and MIPC levels but decreased M(IP)_2_C levels [90]. This would suggest that SNF1 could either directly or indirectly activate the inositol phosphotransferase Ipt1. Moreover, the constitutive active *snf1^G53R^* mutant was shown to rescue the nitrogen starvation-induced cell death of a strain lacking Csg2, an enzyme required for mannosylation of IPC to produce MIPC [91]. Given that both Pho85 and SNF1 phosphorylate and fine-tune the TORC1-dependent activation of Sch9, it would be interesting to further analyze their interplay with respect to the metabolism of sphingolipids.

## 4. The Role of Sch9 in Metabolic Reprogramming and Stress Responses

As TORC1 and Sch9 are central players in the nutrient-controlled signaling network of yeast, it is not surprising that they have a crucial role in controlling growth, stress responses, and longevity (Figure 2). For instance, to support exponential fermentative growth, the TORC1-Sch9 pathway cooperates with the Ras-cAMP-PKA pathway to enhance protein synthesis by boosting the translation capacity of yeast cells. As such, both Sch9 and PKA stimulate ribosome biogenesis by hyperphosphorylating and influencing the subcellular localization of the transcription repressors Stb3, Dot6, and Tod6 [9,92,93] and the RNA polymerase III repressor Maf1 [94]. In addition, both kinases have a significant impact on cell cycle progression through the regulation of different downstream effectors, such as the ubiquitin-conjugating enzyme Cdc34, which controls the degradation of cyclins and cyclin-dependent kinase inhibitors [95]. Note that Sch9 was recently shown to be directly targeted by at least two cyclin-dependent kinases i.e., Pho85 and Bur1 [26,27], and these may provide cell cycle-dependent feedback given their roles in the elongation of telomeres [96,97].

TORC1 and Sch9 are intimately linked to the metabolic reprogramming during the diauxic shift transition and the proper entry of yeast cells into the non-dividing quiescent state (G_0_) [98]. Here, the highly conserved energy sensor SNF1 plays an opposing role to TORC1 by promoting stress responses [99]. In order to guarantee energy homeostasis, SNF1 tunes down TORC1 activity, particularly during glucose starvation [100,101]. In our recent study [28], we performed a SNF1 phosphoproteomic analysis, which allowed us to identify direct SNF1 substrates. This demonstrated that SNF1 not only acts directly on the TORC1 complex itself, as previously shown [102], but that SNF1 also directly phosphorylates Sch9-Ser^288^, thereby contributing to the inhibition of the TORC1 signaling pathway [28]. In addition, recent studies confirmed that reduced TORC1 activity drives cells into the quiescent state by unlocking signaling by several kinases, including Atg1, Gcn2, Npr1, Rim15, Yak1, and Mpk1/Slt2 [34,35,103]. In connection to the TORC1-Sch9 axis, previous data revealed that it cooperates with the Ras-cAMP-PKA pathway to control the cytoplasmic sequestration of the Greatwall protein kinase Rim15 via association with the 14-3-3 protein Bmh2, thereby preventing its activation when nutrients are plentifully available [12,104,105,106]. Importantly, the TORC1-Sch9 axis is itself a regulator of PKA activity as it prevents the phosphorylation and activation of Bcy1, the negative regulatory subunit of PKA, via the cell wall integrity MAPK Mpk1/Slt2 [31]. Moreover, Sch9 also indirectly controls, through Yak1 and the retention factor Zds1, the carbon source-dependent nucleocytoplasmic distribution of Bcy1, the stability and nucleocytoplasmic distribution of the PKA catalytic subunit Tpk2, and regulates the phosphorylation of the Ras GAP Cdc25 [107,108]. It is intriguing that Mpk1/Slt2 was also reported to inhibit TORC1 under conditions of ER stress [109] as it raises the possibility that Sch9 may also provide feedback to TORC1 via the MAPK. Once at the diauxic shift, the inhibitory actions of TORC1-Sch9 and PKA on Rim15 are relieved, and the kinase translocates into the nucleus to activate such transcription factors as Msn2/4 and Hsf1, thereby inducing the expression of genes that contain a stress-responsive or heat shock-responsive element in their promoter, respectively [12,104,110]. In addition, Rim15 indirectly controls the activity of the transcription factor Gis1 via the endosulfines Igo1/2 and PP2A-Cdc55 phosphatase and, as such, allows for the induction of genes containing the post-diauxic shift promoter element [111,112,113,114]. Interestingly, the nuclear retention of Rim15 is regulated by the Pho85-Pho80 CDK-cyclin pair, which phosphorylates Rim15 to dictate its nuclear export [115,116]. Hence, Pho85-Pho80 maintains a feed-forward system since, besides priming Sch9 for full activation by TORC1 [26], the CDK-cyclin pair directly controls the nucleocytoplasmic translocation of Rim15 to adjust the execution of the Rim15-dependent G_0_ program response to phosphate availability. In parallel to Rim15, TORC1 and PKA signaling similarly maintain inactive Yak1 in the cytoplasm by tethering this kinase to the cytoplasmic 14-3-3 anchor proteins Bmh1/2 [117,118]. Once this inhibition is relieved at the diauxic shift, Yak1 becomes nuclear and impacts Msn2/4-, Hsf1-, and Gis1-mediated transcription as well [119,120]. Here, Sch9 plays a dual role since one study suggested that Sch9 not only phosphorylates Yak1 but that it also controls the stability of Yak1 during growth and stationary phase [108]. Notably, glucose starvation further stimulates Yak1 to phosphorylate the Ccr4-Not subunit Pop2, which is essential to arrest the cell cycle at G_1_ and to ensure proper entry into the stationary phase [121]. Yak1 also phosphorylates the transcriptional co-repressor Crf1, which inhibits the transcription of ribosomal genes [122]. Moreover, consistent with its role as a potential Sch9 substrate, Crf1 controls the nucleocytoplasmic distribution of Bcy1 as well [123]. Besides Rim15 and Yak1, there is a third kinase that has been proposed to act downstream of the TORC1-Sch9 axis for the entry into the quiescent state, namely, the GSK-3 homolog Mck1 [34,124]. Mck1 was originally identified as a downstream effector of the Pkc1-Mpk1/Slt2 cell wall integrity pathway that affects the subcellular redistribution of Bcy1 in response to heat stress [125,126]. Given the involvement of Mpk1/Slt2, it is not surprising that later studies confirmed Mck1 to be under negative control of the TORC1-Sch9 axis to coordinate reserve carbohydrates metabolism [127], the repression of ribosomes and tRNA synthesis [128], and the expression of different stress-induced and post-diauxic genes [129,130,131] in response to nutrient limitation. Moreover, this link between Mck1 and Sch9 is further corroborated by the observation that loss-of-function mutations or deletion of *MCK1* partially suppress the growth defects of *sch9Δ* cells under fermentative and respiratory conditions [132]. In line with these observations, our own phosphoproteomic analysis suggested that TORC1 inhibits the quiescence program in part via the Sch9-dependent inhibition of Mck1 [34].

Finally, SNF1 also has a profound impact on stress responses. Indeed, Msn2/4 and Gis1 were originally retrieved as multicopy suppressors of SNF1 defects [133,134], and although active SNF1 acts as an inhibitor of Sch9 [28], it also directly phosphorylates Msn2 and Hsf1, thereby constraining the nuclear localization of these factors and adapting the transcriptional stress response during glucose starvation [135,136,137]. Notably, SNF1 is also a negative regulator of PKA since it phosphorylates adenylate cyclase, thereby lowering cAMP levels [138].

## 5. The Involvement of Sch9 in Proteasomal Degradation and Autophagy

For optimal growth, the timely degradation of proteins to maintain proteostasis is essential. Different studies indicate that protein degradation is also used by yeast cells to adjust metabolic programming and fine-tune stress responses. For instance, TORC1 signaling is known to control the multivesicular body (MVB) pathway-driven degradation of plasma membrane proteins and lipids [139,140,141,142]. In addition, several transcription factors are known to be under the control of TORC1, such as Gcn4, Gln3, or Gat1, involved in amino acid biosynthesis and nitrogen catabolite repression, but also the stress-responsive factors Msn2/4, Gis1, and Hsf1, have all been shown to be subject of proteasomal degradation, thereby leading to the adjustment of their transcriptional responses [120,143,144,145]. In fact, one of these studies demonstrated that the proteasome is not only required to prevent activation of starvation-specific genes during exponential growth, but it is also essential for yeast cells to adapt to reduced TORC1 activity [120]. Interestingly, proteasome abundance and proteasome assembly are themselves regulated by TORC1 signaling [120,146,147,148]. It is likely that Sch9 is also involved since proteasome abundance is managed via the transcription of genes encoding proteasomal subunits through the transcription factor Rpn4, which itself is induced by the Hsf1 [149]. Furthermore, proteasome assembly is regulated by the translation of proteasome regulatory particle assembly chaperones (RPACs), which is under the control of Mpk1/Slt2 [147]. In cooperation with SNF1, Mpk1/Slt2 further controls the formation of proteasome storage granules upon the inhibition of mitochondrial function and the drop in ATP levels following carbon starvation [150,151]. These granules represent reversible cytosolic proteasome condensates that serve to protect cells against stress, as they are believed to shield the proteasome from autophagic degradation or proteophagy, which, besides SNF1 and Mpk1/Slt2, also involves TORC1 [152,153,154]. Upon exit from quiescence and resumption of cell proliferation, the proteasome storage granules rapidly resolve, and proteasomes reenter the nucleus [150].

As the master regulator controlling cell growth and metabolic activity, TORC1 plays a central role in the regulation of autophagy. Under nutrient-rich conditions, TORC1 is active, and general autophagy is inhibited. This inhibition is accomplished by the TORC1 pool localized at perivacuolar signaling endosomes and involves the phosphorylation of Atg13 to prevent its association with Atg1 and, thereby, the induction of macroautophagy [56,155,156]. In addition, TORC1 phosphorylates the Vps27 subunit of ESCRT-0 to antagonize cargo selection for microautophagy and the turnover of vacuolar membrane-resident and associated proteins through direct engulfment by the vacuolar membrane [56,157,158]. Interestingly, the trafficking of Vps27 to the vacuole and ESCRT-dependent microautophagy are also controlled by SNF1 under glucose-limiting conditions [159]. Whether Sch9 has a role in microautophagy is currently unknown. Nonetheless, Sch9 is clearly important for the inhibition of macroautophagy under nutrient-rich conditions, as the combined inactivation of Sch9 and PKA induces macroautophagy through a process that requires the Atg1-Atg13-Atg17 complex, Rim15, and Msn2/4 [14]. Consistent with this observation, Atg13 is also phosphorylated by PKA at residues that are distinct from those targeted by TORC1 [160]. A more recent study suggested that the induction of bulk autophagy at the diauxic shift occurs mainly via the inactivation of PKA and Sch9 and established that this is mediated by the cell wall integrity sensor Mtl1, which signals glucose limitation to Ras2 and Sch9 [161]. This pathway also controls the autophagic degradation of mitochondria when cells reach the stationary phase [161]. Indeed, most recent findings demonstrated that this type of glucose starvation-induced autophagy requires the recruitment of the DNA-damage sensor Mec1 to mitochondria, where it is phosphorylated by SNF1 and where it binds Atg1 and Atg13 to associate with the phagophore assembly site [162,163]. As for the roles of Rim15 and Msn2/4, several studies indicated that they control the transcription of several autophagy genes. Rim15 impacts on the expression of different *ATG* genes by relieving repression mediated by the histone demethylase protein Rph1 and the Ume6-Sin3-Rpd3 histone deacetylase complex [164,165,166]. Msn2/4, on the other hand, activates transcription of *ATG* genes, as shown, for instance, for *ATG8* [167] and *ATG39* [168]. Importantly, TORC1 signaling additionally controls the expression of *ATG* genes via the transcription factors Gln3, Gat1, and Gcn4 [166].

In contrast to SNF1, which inhibits both TORC1 and Sch9 once activated under nutrient-limiting conditions [28,102,169], the Pho85-Pho80 CDK-cyclin pair boosts Sch9 activity under nutrient-rich conditions by enhancing PI(3,5)P_2_ production and by priming Sch9 for its subsequent activation by TORC1 [26]. In addition, Pho85-Pho80, either directly or indirectly, enhances Atg13 phosphorylation [26] and antagonizes the nuclear accumulation of Rim15 under glucose-limiting conditions [115,116]. Hence, it is not surprising that Pho85-pho80 has been reported to be a negative regulator of autophagy [170,171]. However, the role of Pho85 in autophagy is more complex. Accordingly, other cyclins are involved as well, and Pho80, Clg1, and Pcl1 combined also positively control autophagy by promoting the degradation of Sic1, a cyclin-dependent kinase inhibitor involved in cell cycle regulation that seemingly also acts as a negative regulator of autophagy by targeting Rim15 [171].

## 6. The Role of Sch9 on Longevity Modulation

Sch9, as part of the TORC1 pathway, is a prime determinant of cellular aging. Loss of Sch9 function increases the survival of stationary non-dividing cells, i.e., chronological lifespan (CLS) [36], and enhances replicative lifespan (RLS), i.e., the number of daughters that a single mother cell can produce asexually [172,173]. Interestingly, Sch9 has also been reported to play a major role in pro-longevity effects promoted by caloric and dietary restriction [174,175,176,177,178]. The role of Sch9 on longevity seems to be more complex and diverse than previously anticipated, being implicated in different stress responses, including the interplay between oxidative and metabolic stresses.

Caloric restriction (CR), characterized by a 10–30% reduction in calories compared to an *ad libitum* diet, is a potent modulator of longevity in several species. Although the longevity mechanisms of CR are not completely uncovered, it is clear that its benefits are related to alterations in the metabolic rate and the accumulation of reactive oxygen species (ROS). Initial studies have implicated inhibition of the TORC1-Sch9 axis as the longevity pathway through which CR modulates lifespan. In RLS, CR-induced longevity is mediated by reduced signaling through TORC1, Sch9, and PKA, which results in the downregulation of ribosome biogenesis. This proposed model for CR effects during RLS is independent of sirtuin 2 (Sir2) but likely links the signaling network from nutrients to ribosome assembly and protein synthesis [173]. Regarding CLS, it was found that CR still promotes CLS extension in cells lacking *SCH9*, suggesting that inhibition of the TORC1-Sch9 axis represents only one of the mechanisms through which CR modulates the lifespan [179]. Downregulation of Sch9, as well as downregulation of Ras2, delays aging through pathways that only partially overlap with the CR-mediated extension of the lifespan [180,181]. CR and Sch9-mediated longevity share the common downstream target Rim15 [36]. In the Sch9-mediated longevity pathway, Rim15 acts, in part, through the stress response transcription factor Gis1 [182], which binds post-diauxic shift elements found in the promoters of genes of the stress-resistance systems, such as *HSP26*, *HSP12*, and *SOD2* [183]. Consistent with this, the deletion of *SOD2* abolishes lifespan extension in *sch9Δ* mutant [184].

In natural scenarios, yeast and other organisms experience periods of nutritional stress, and an appropriate and efficient metabolic adaptation is, therefore, essential to ensure cell survival. Sch9 is a key player in this metabolic adaptation and in the assembled response to nutrient availability. Sch9 receives many different inputs and executes its function accordingly. As introduced above, one of the most important upstream kinases is the TORC1 kinase that targets Sch9 under favorable growth conditions [8,185], but Sch9 also has TORC1-independent functions under multiple stress conditions. In agreement with this notion, it has been shown that reducing Sch9 activity extends lifespan when yeast cells are pre-grown under nutrient-rich conditions, but it shortens the lifespan when pre-grown under nutrient-poor conditions [186]. This indicates that pre-adaptations to respiratory metabolism and oxidative stress play a central role in determining cellular longevity. As such, the deletion of both *TOR1* and *SCH9* results in the increased respiratory capacity associated with a higher ratio of mitochondrial respiratory-chain enzymes per mass during active growth [187,188]. Although the underlying mechanisms are still poorly understood, the transcription factors Hcm1 and Hap4 have been implicated in the nuclear regulation of mitochondrial respiration. While Sch9 directly phosphorylates Hcm1 to inhibit its nuclear import, it indirectly regulates Hap4 through sphingolipids signaling [76,189].

The elevated respiratory capacity of *tor1Δsch9Δ* cells is, therefore, associated with increased ROS levels and increased longevity by a hormesis-like phenomenon [190]. The so-called hormesis effect mediates lifespan extension by an adaptative mitochondrial ROS signaling that, even under CR conditions, is independent of Rim15, which, as mentioned above, is a well-known target of Sch9 [191,191,192]. Nevertheless, when ROS levels are above a certain threshold, mitochondria can cause irreversible cellular damage, triggering regulated cell death and premature aging. This dichotomous function of mitochondria indicates that, as in other organisms, mitochondrial function is also a relevant hallmark of aging in yeast.

The TORC1-independent function of Sch9 on longevity seems to be mainly related to the activation of specific gene promoters and transcription factors. For example, as mentioned above, Sch9 regulates oxidative stress response by indirectly acting on the transcription factor Gis1 [182]. Sch9 can also crosstalk with the Hog1 MAP kinase via the Sko1 transcription factor, which activates stress gene expression upon a high osmolarity [8,185]. In addition to activating transcription factors, Sch9 can also affect gene expression through chromatin remodeling. Sch9 is required for the phosphorylation of the residue Thr^11^ of histone H3 under stress, and the loss of pThr^11^ prolongs CLS by altering the stress response at an early stage of the CLS [193]. Therefore, Sch9 also links nutritional stress to chromatin remodeling during aging.

Deletion of the *SCH9* also significantly decreases the overall mutation frequency and DNA damage during CLS [194], resulting in reduced genomic instability [179,195]. The increased *SOD2* expression and, consequently, reduced superoxide-induced DNA damage found in *sch9Δ* cells further contribute to the observed reduced genomic instability.

As previously referred, Sch9 acts on cell cycle regulation by promoting an efficient G1 arrest [196]. By inhibiting Rim15, Sch9 promotes the proteolysis of Sic1, a CDK inhibitor [95,196,197]. In accordance, the deletion of *SIC1* results in S phase entry and reduction in CLS by increased superoxide generation [198,199], and the constitutive activation of Sch9 shortens the CLS by a defect in proper G_1_ arrest [95]. Importantly, the deletion of *SCH9*, but not CR, protects against the premature aging phenotype of yeast cells lacking the RecQ helicase Sgs1 (WRN and BLM homolog) by inhibiting error-prone recombination and preventing DNA damage and dedifferentiation [194]. It appears that enhanced cellular protection against stress, tighter G1 arrest, and reduced recombination errors are mechanisms by which the lack of Sch9 activity protects cells against genomic instability and dedifferentiation associated with accelerated aging when the Sgs1 is mutated [194].

Yeast aging is likely a suitable natural scenario to understand the role of Sch9 in the interplay between different stress responses and nutritional status, which can constitute, per se, a particular form of stress. Because of their differential participation in yeast longevity, the two main yeast aging models can provide valuable information regarding the signal transduction mediated by Sch9 in response to multiple inputs.

## 7. Conclusions and Future Directions

The investigation of the topology, regulation, and crosstalk of signaling pathways has proven invaluable in the dissection of the molecular mechanisms underlying pathophysiological processes [200,201,202]. During evolution, a rewiring of signaling cascades often occurred, reflecting the different environment of organisms, their uni- or multicellularity, and specialization. However, the use of regulation hubs, which integrate upstream inputs to conveniently control multiple downstream effectors at once, is a remarkably conserved system [203]. In this context, the yeast kinase Sch9 exemplifies the central node of a so-called bow-tie signaling network (Figure 2) [204].

In spite of the extensive studies which started to unravel the complex regulation and function of Sch9 and that are reviewed in depth above, recent findings hint at currently overlooked regulatory features. For instance, Sch9 was found to be phosphorylated by the CDK Bur1 at the Ser^560^, Thr^568^, Thr^574^, Thr^575^, Ser^709^, Thr^710^, Ser^711^, and Thr^721^ residues, in addition to the more deeply studied Thr^723^ and Ser^726^ residues [27]. Furthermore, at least 12 lysine residues of Sch9 were reported to be ubiquitinated in high-throughput studies [205,206], but the regulation of the possible conditional Sch9 ubiquitination and degradation has not been investigated to date. Thus, future studies focused on these post-translational modifications could pinpoint new Sch9 regulatory mechanisms.

Sch9 does not have a readily identifiable orthologue in mammals. However, it shares functional similarities with the mammalian AGC kinase family member AKT (also known as protein kinase B or PKB), which exists in three different isoforms, AKT1, AKT2, and AKT3 [207], in what concerns its role in cellular signaling and regulation of growth-related processes [208]. Notably, Sch9 has also been suggested to be functionally related to the homologous mammalian S6K, although it appears that the cellular roles of yeast Ypk3 may more closely overlap with the ones of S6K in mammals [209,210]. Based on our current knowledge, we deem it possible that the functional similarities between Sch9 and AKT may also be extended to the role of these kinases in regulating longevity. In this context, the role of AKT in longevity is complex and multifaceted, ranging from promoting cell growth and survival, which can have beneficial effects on tissue repair and maintenance that are important for healthy aging, to detrimental effects that are related to increased cellular senescence or cancer promotion (when AKT is hyperactive) [200]. Studies on Sch9 could, thus, help elucidate the evolutionary functional origin of AKT and shed light on some of its most important functions. While the direct translation of findings from Sch9 in yeast to AKT in mammals may not be straightforward, studying the conserved functional aspects of these pathways can provide valuable information and generate hypotheses for further investigation in mammalian systems. Even though, as abovementioned and extensively presented in the current review, several features of the regulation and the regulators of Sch9 suggest some degree of functional homology to AKT in higher eukaryotes, remarkable differences also highlight the divergence between these two important signaling effectors. For instance, the lack of PI(3,4,5)P_3_ in budding yeast [211] and the absence of a PH domain in Sch9 (Table 1) prevent its localization to the plasma membrane, the subcellular localization where AKT is activated and carries out its most studied functions [212].

Thus, it is important to consider the limitations and differences between yeast and mammalian biology, but utilizing Sch9 as a model can still contribute to our understanding of the broader principles and mechanisms involved in the regulation of pathophysiological processes. Specifically, investigation of the post-translational modifications and their impact on the structure and function of Sch9 may translate to equivalent findings regarding the biochemical regulation of AKT and related protein kinases. Moreover, as examined in depth in this review, Sch9 could serve as a case study for lipid-dependent regulation of signaling effectors, as well as a molecular tool to investigate the subcellular distribution of specific lipid species [16]. Finally, Sch9 shares upstream regulators with other AKT-like proteins [19,24,86], and it is thus likely that Sch9 regulation studies could help expand the understanding of signaling networks of cognate kinases.

## Figures and Tables

**Figure 1 jof-09-00787-f001:**
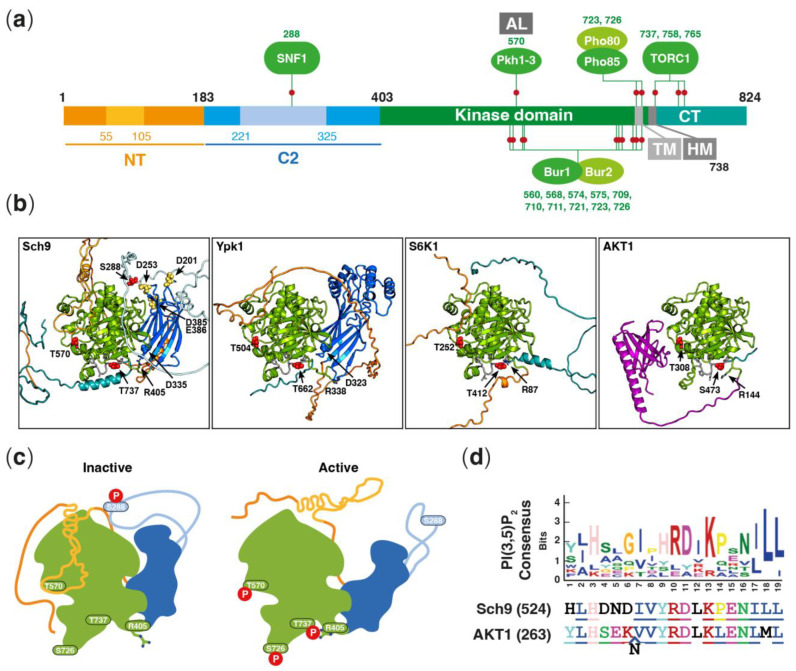
Structural properties of Sch9. (**a**) Schematic domain architecture of Sch9. Red dots represent residues that are phosphorylated by the indicated protein kinases (green numbers refer to the phospho-residues within Sch9). NT, N-terminus; C2, C2 domain; AL, activation loop; TM, turn motif; HM, hydrophobic motif; CT, C-terminus. (**b**) Side-by-side comparison of the Alphafold2 [39,40] predicted structures of Sch9 (AF-P11792-F1_v4.pdb), Ypk1 (AF-P12688-F1_v4.pdb), S6K1 (AF-P23443-F1_v4.pdb), and AKT1 (AF-P31749-F1_v4.pdb). The structural information is represented in cartoon style, and the domains (see Table 1) are colored as follows: N-terminus in orange; pleckstrin homology (PH) + linker in purple; C2 domain in blue; kinase domain in green; hydrophobic motif (HM) in grey (with side chains); and C-terminus in teal. The labeled phosphosites of the activation loop, the HM, and the C2 extended loop are displayed as balls and colored in red. The amino acids that could potentially coordinate Ca^2+^ on the top of the C2 domain of Sch9 are labeled, displayed as balls, and colored in yellow. The arginines corresponding to the Arg^144^ residue of AKT1 are labeled, and the side chains are represented as sticks. (**c**) Model depicting the spatial relationship of the kinase domain with the N-terminal region and the C2 domain and kinase domain of Sch9 in its inactive and active configuration, including the steric occlusion of the active site by the N-terminus in inactive Sch9 and the electrostatic interaction between Arg^405^ and pThr^737^ in the active configuration. (**d**) Proposed consensus sequence for PI(3,5)P_2_ binding [41] and the corresponding motifs upstream of the activation loop in Sch9 and AKT1.

**Figure 2 jof-09-00787-f002:**
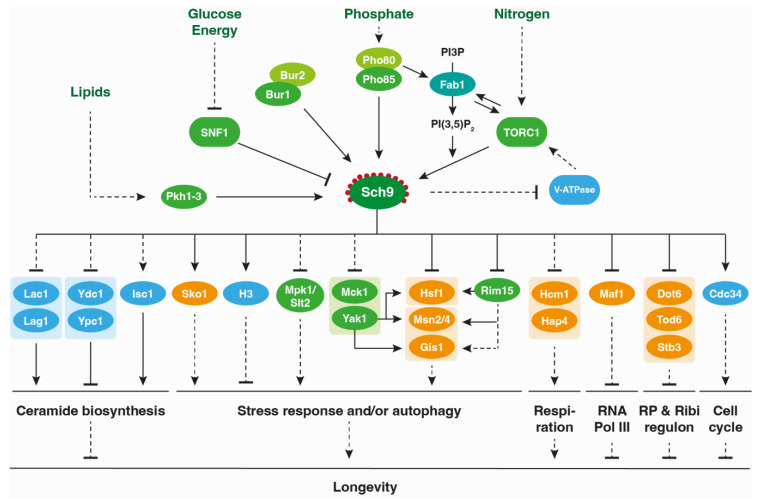
Model of the Sch9 signaling network. Protein kinases, transcription factors, and various other proteins are colored green, orange, and blue, respectively. The lipid kinase Fab1 is colored turquoise. Arrows and bars refer to direct (full line) or indirect (dashed line) activating and inhibitory interactions, respectively. See main text for more details.

**Table 1 jof-09-00787-t001:** Structural domains of Sch9 and close homologues.

	Yeast		Mammalian	
Region	Sch9	Ypk1	S6K1	AKT1
N-terminus	1–183	1–117	1–83	See PH domain
PH + Linker	-	-	-	1–141
C2 domain	184–402	118–336	-	-
C2 extended loop	221–325	-	-	-
Kinase domain	403–738	337–663	84–413	142–474
Activation loop	570–574 (TFCGT)	504–508 (TFCGT)	252–256 (TFCGT)	308–312 (TFCGT)
Hydrophobic motif	733–738 (FAGFTF)	658–663 (FGGWTY)	408–413 (FLGFTY)	469–474 (FPQFSY)
C-terminus	739–824	664–680	414–525	475–480

## Data Availability

Data sharing is not applicable to this article as no datasets were generated or analyzed in the course of the current study.

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
