# Peer review of "The Yeast Protein Kinase Sch9 Functions as a Central Nutrient-Responsive Hub That Calibrates Metabolic and Stress-Related Responses"

_jof, 2023, doi:10.3390/jof9080787_

Round 1

Reviewer 1 Report

The authors have presented a comprehensive review of the kinase Sch9, discussing its structural characteristics, regulations, and cellular mechanisms in which Sch9 plays a role. The content of the review holds relevance to the field, and there is a lack of similar publications.

Minor revisions:

In the introduction, you mention the kinases Pkh1, Pkh2, and Pkh3. It would be beneficial to provide information on the signal transduction pathways in which these kinases participate or the specific cellular processes they are involved in. While you provide more details about other kinases in the introduction, these specific details are missing for Pkh1, Pkh2, and Pkh3. You make a reference to the phosphorylation of Thr570. Could you please elaborate on the specific factors or conditions that led to the phosphorylation of this particular residue?

In relation to the role of Sch9 in longevity, you discuss its involvement under caloric restriction conditions. Several studies, such as Lyu et al. 2019 (PMID: 31582588), Picazo et al. 2015 (PMID: 25658705), Santos et al. 2015 (PMID: 25576917), Mirisola et al. 2014 (PMID: 24516402), and Wu et al. 2013 (PMID: 23691220), have also mentioned the involvement of Sch9 in dietary restriction. It would be worthwhile to incorporate these findings as it represents another phenomenon where Sch9 is implicated.

Author Response

In the introduction, you mention the kinases Pkh1, Pkh2, and Pkh3. It would be beneficial to provide information on the signal transduction pathways in which these kinases participate or the specific cellular processes they are involved in. While you provide more details about other kinases in the introduction, these specific details are missing for Pkh1, Pkh2, and Pkh3. You make a reference to the phosphorylation of Thr570. Could you please elaborate on the specific factors or conditions that led to the phosphorylation of this particular residue?

We thank the reviewer for their comment, and we have amended the manuscript accordingly. In the Introduction, we added some details about Pkh1/2/3 activation: “These include the paralogous Pkh1, Pkh2, and Pkh3 kinases, which themselves are stimulated (Liu et al. 2005. J. Biol. Chem., Friant et al. 2001. EMBO J., Luo et al. 2008. J. Biol. Chem.), or not (Roelants et al. 2010. PNAS) by phytosphingosine (PHS) (see Roelants et al. 2017. Biomolecules for a detailed discussion), and which phosphorylate Thr570 in the activation loop of Sch9 (Urban et al. 2007. Mol. Cell., Liu et al. 2005. J. Biol. Chem., Roelants et al. 2004. Microbiol. Read. Engl., Voordeckers et al. 2011. J. Biol. Chem.).”.

Moreover, in the Lipid-dependent regulation of Sch9 section, we added: “Here, a key role is played by the protein kinases Pkh1, Pkh2, and Pkh3, the orthologues of the mammalian 3-phosphoinositide dependent kinase-1, PDK1, which are known to be involved in maintenance of cell wall integrity and the control of eisosome dynamics (Luo et al. 2008. J. Biol. Chem., Roelants et al. 2017. Biomolecules, Roelants et al. 2002. Mol. Biol. Cell). These PDK1 orthologs phosphorylate Sch9 in its activation loop at Thr570 (Figure 1a), an event that occurs independently of the C-terminal Sch9 phosphorylation by TORC1 and that is required to obtain full Sch9 activity (Urban et al. 2007. Mol. Cell., Voordeckers et al. 2011. J. Biol. Chem.).”.

In relation to the role of Sch9 in longevity, you discuss its involvement under caloric restriction conditions. Several studies, such as Lyu et al. 2019 (PMID: 31582588), Picazo et al. 2015 (PMID: 25658705), Santos et al. 2015 (PMID: 25576917), Mirisola et al. 2014 (PMID: 24516402), and Wu et al. 2013 (PMID: 23691220), have also mentioned the involvement of Sch9 in dietary restriction. It would be worthwhile to incorporate these findings as it represents another phenomenon where Sch9 is implicated.

We thank the reviewer for their comment. Although we could not discuss in detail all the single findings of the above-mentioned studies, we included in the corresponding section a reference to these studies: ”Interestingly, Sch9 has also been reported to play a major role in pro-longevity effects promoted by caloric restriction (Lyu et al. 2019. Aging, Picazo et al. 2015. PLoS One, Santos et al. 2015. Oncotarget, Mirisola et al. 2014. PLoS Genet., and Wu et al. 2013. PLoS One.”.

Reviewer 2 Report

The review by Caligaris et al., is a wonderful review of the current literature both for those in the field and a more general audience. Much appreciated was the fact that the authors distinguished what is known, from what is likely to be true, from what is speculated or possible. The work sets up many hypotheses for the authors’ groups or those of others to pursue thereby testing their ideas and/or providing the currently missing information and connecting the dots.

There is, however, one major deficiency in the work that significantly diminishes its value especially to a general audience. The authors should construct more figures like Figure 2 such that all of the genes and proteins they discuss are represented in figures as well as the text. Different types of arrows and bars could distinguish what is known from what is likely to be true from what is speculated or possible. As the piece currently exists, a general reader will quickly lose interest because of the inherent complexity of the regulatory circuits being described. The fear of lending greater credence to one model/possibility or another can be delt with/mitigated by clearly stating the meaning of the symbols in a manner analogous to what was done in the Figure 2 legend. I recognize that this will involve multiple symbols. That is OK. Readers require visual assistance as the material becomes more complex and convoluted. This will not be an easy task for the authors but will greatly enhance the quality and acceptance of their work by the general fungal community. 

Below are a list of suggestions that will make the manuscript more readable.

 Line 41. “yeast. It consist of”

 Line 150. Replace into” with “to” and delete “a”

 175-177. Split into two sentences.

 Line 178. Change “part” to “region”

 Lines 190-191. “this too involves”

 Line 202 pathways, such as Pho85

 Lines 207-208.

 Line 216. Change impact to influence

 Lines 220-223. Break into two sentences.

 Line 230  played by protein kinases, delete the

 Line 232. Delete “in its” and “at”

 Line 249. Delete the period after cells.

 Lines 268-271. Break into two sentences

 Line 314. Delete “on the residue”

 Line 316. “ulocking signaling”

 Line 320. Bmh2, thereby preventing

 Line 320. plentiful delete ly and available

 Line 326. May also provide

 Line 327. Delete “as well”

 Line 338 Change “function of” to “response to”

 Line 343. Need a better word than “particular”

Line 343 delete not only

 Line 344. “not only phosphorylates”

 Lines 376-380. Break into two sentences.

 Lines 385-388. Break into two sentences.

Lines 414-419. Break into two sentences.

Author Response

The review by Caligaris et al., is a wonderful review of the current literature both for those in the field and a more general audience. Much appreciated was the fact that the authors distinguished what is known, from what is likely to be true, from what is speculated or possible. The work sets up many hypotheses for the authors’ groups or those of others to pursue thereby testing their ideas and/or providing the currently missing information and connecting the dots.

We warmly thank the reviewer for their appreciation of our manuscript. The paper was indeed conceived not only as a collection of published literature but also as an opportunity to make some hypotheses to guide further investigation in the field.

There is, however, one major deficiency in the work that significantly diminishes its value especially to a general audience. The authors should construct more figures like Figure 2 such that all of the genes and proteins they discuss are represented in figures as well as the text. Different types of arrows and bars could distinguish what is known from what is likely to be true from what is speculated or possible. As the piece currently exists, a general reader will quickly lose interest because of the inherent complexity of the regulatory circuits being described. The fear of lending greater credence to one model/possibility or another can be delt with/mitigated by clearly stating the meaning of the symbols in a manner analogous to what was done in the Figure 2 legend. I recognize that this will involve multiple symbols. That is OK. Readers require visual assistance as the material becomes more complex and convoluted. This will not be an easy task for the authors but will greatly enhance the quality and acceptance of their work by the general fungal community.  

We agree with the reviewer that the graphic representation of concepts greatly help the readers. However, we decided to not add further figures for various reasons: the review is intended to be focused on Sch9, and that alone took a relevant amount of space; almost all involved players are already depicted in Figure 2; for all the remaining proteins/pathways, the relevant references are mentioned in the text. We would like to add that other reviewers did not ask for additional figures but asked for clarifications concerning some of the pathways (e.g.: Pkh1-3 kinases), which can now be found in the revised manuscript.

Below are a list of suggestions that will make the manuscript more readable.

We thank the reviewer for their editing suggestions, which we almost entirely accepted and introduced in the manuscript.

Reviewer 3 Report

The paper of Caligaris et al is an interesting Review on the functions of Sch9 yeast protein kinase. This review is up to date and enriched by a huge reference list and it is focused on  the structural properties of Sch9 protein well described in Figure 1 and in the lipid dependent regulation of its activity. Moreover the role of Sch9 in stress response and in the longevity is also well represented and modeled in Fig 2.

Since SCH9 gene was discovered as a multicopy suppressor of  growth arrest caused by a downregulation of Cdc25/Ras/PKA pathway (Toda, T., Cameron, S., Sass, P., and Wigler, M. (1988) SCH9, a gene of Saccharomyces cerevisiae that encodes a protein distinct from, but functionally and structurally related to, cAMP-dependent protein kinase
catalytic subunits. Genes Dev. 2, 517–527) and there is an huge literature on the complex interaction between Sch9 and PKA pathways it is surprising that in this review this particular aspect is only marginally described, for example the interaction between Sch9 and Bcy1 could be better commented and analyzed.
Another point that could be considered is the recent availability of specific tools to measure "in vivo" the activities of PKA and Sch9 (Plank et al., 2020, Molecular & Cellular Proteomics 19, 655–671; Colombo et al Cell Signal. 2022 Apr; 92: 110262; Botman et al. FEMS Yeast Res. 2023 Jan 4;23:foad029. doi: 10.1093/femsyr/foad029.) that could be useful to evaluate the activities of these kinases also in single yeast cells.

Author Response

Since SCH9 gene was discovered as a multicopy suppressor of  growth arrest caused by a downregulation of Cdc25/Ras/PKA pathway (Toda, T., Cameron, S., Sass, P., and Wigler, M. (1988) SCH9, a gene of Saccharomyces cerevisiae that encodes a protein distinct from, but functionally and structurally related to, cAMP-dependent protein kinase
catalytic subunits. Genes Dev. 2, 517–527) and there is an huge literature on the complex interaction between Sch9 and PKA pathways it is surprising that in this review this particular aspect is only marginally described, for example the interaction between Sch9 and Bcy1 could be better commented and analyzed.

We thank the reviewer for their comment. We agree that in the manuscript we overlooked the origin of SCH9 discovery. Accordingly, we have added a sentence in the Introduction: “SCH9 (for Scott Cameron HindIII library clone number 9; Takashi Toda; personal communication) has been initially identified as a multicopy suppressor of the growth defect caused by a temperature-sensitive cdc25 allele (Toda et al. 1988. Genes Dev.), hinting at a connection with the Ras/PKA signaling pathway.”

As for the functional interaction between Sch9 and PKA, we genuinely think that we cited all the relevant literature. However, agreeing with the reviewer that some concept could have been written more explicitly, we added the following sentence: “Moreover, Sch9 also indirectly controls through Yak1 and the retention factor Zds1 the carbon source-dependent nucleocytoplasmic distribution of Bcy1, the stability and nucleocytoplasmic distribution of the PKA catalytic subunit Tpk2 and regulates the phosphorylation of the Ras GAP Cdc25 (Zhang et al. 2011. FEBS Lett., Zhang and Gao 2012. FEMS Microbiol Lett.)”.

Another point that could be considered is the recent availability of specific tools to measure "in vivo" the activities of PKA and Sch9 (Plank et al., 2020, Molecular & Cellular Proteomics 19, 655–671; Colombo et al Cell Signal. 2022 Apr; 92: 110262; Botman et al. FEMS Yeast Res. 2023 Jan 4;23:foad029. doi: 10.1093/femsyr/foad029.) that could be useful to evaluate the activities of these kinases also in single yeast cells.

We thank the reviewer for their comment. However, due to space constraints and the aim of the review, we decided not include in the manuscript details about the technical challenges to evaluate the activity of Sch9 or other AGC kinases.